# Interference of Celastrol with Cell Wall Synthesis and Biofilm Formation in *Staphylococcus epidermidis*

**DOI:** 10.3390/antibiotics14010026

**Published:** 2025-01-03

**Authors:** Leandro de León Guerra, Nayely Padilla Montaño, Laila Moujir

**Affiliations:** 1Departamento de Bioquímica, Microbiología, Biología Celular y Genética, Facultad de Farmacia, Universidad de La Laguna, Avenida Astrofísico Fco Sánchez s/n, 38206 La Laguna, Spainnayely.padilla@correo.uady.mx (N.P.M.); 2Laboratorio de Farmacología, Facultad de Química, Universidad Autónoma de Yucatán, Calle 43 s/n x 96, Paseo de las Fuentes y 40 Col, Mérida 97069, Mexico

**Keywords:** celastrol, *Staphylococcus epidermidis*, mechanism of action, biofilm

## Abstract

**Background**: The emergence of antibiotic-resistant bacteria, including *Staphylococcus epidermidis*, underscores the need for novel antimicrobial agents. Celastrol, a natural compound derived from the plants of the Celastraceae family, has demonstrated promising antibacterial and antibiofilm properties against various pathogens. **Objectives:** This study aims to evaluate the antibacterial effects, mechanism of action, and antibiofilm activity of celastrol against *S. epidermidis*, an emerging opportunistic pathogen. **Methods**: To investigate the mechanism of action of celastrol, its antibacterial activity was evaluated by determining the time–kill curves, assessing macromolecular synthesis, and analysing its impact on the stability and functionality of the bacterial cell membrane. Additionally, its effect on biofilm formation and disruption was examined. **Results:** Celastrol exhibited significant antibacterial activity with a minimal inhibitory concentration (MIC) of 0.31 μg/mL and minimal bactericidal concentration (MBC) of 15 μg/mL, which is superior to conventional antibiotics used as control. Time–kill assays revealed a concentration-dependent bactericidal effect, with a shift from bacteriostatic activity at lower concentrations to bactericidal and lytic effect at higher concentrations. Celastrol inhibited cell wall biosynthesis by blocking the incorporation of N-acetylglucosamine (NAG) into peptidoglycan. In contrast, the cytoplasmic membrane was only affected at higher concentrations of the compound or after prolonged exposure times. Additionally, celastrol was able to disrupt biofilm formation at concentrations of 0.9 μg/mL and to eradicate pre-formed biofilms at 7.5 μg/mL in *S. epidermidis*. **Conclusions**: Celastrol exhibits significant antibacterial and antibiofilm activities against *S. epidermidis*, with a primary action on cell wall synthesis. Its efficacy in disrupting the formation of biofilms and pre-formed biofilms suggests its potential as a therapeutic agent for infections caused by biofilm-forming *S. epidermidis* resistant to conventional treatments.

## 1. Introduction

*Staphylococcus epidermidis*, a coagulase-negative staphylococci (CoNS), is a ubiquitous member of the human skin and mucosal membranes. This commensal bacterium plays a pivotal role in maintaining skin homeostasis by activating the immune response and inhibiting colonization by opportunistic pathogens [1]. Despite its beneficial role under normal conditions, *S. epidermidis* has emerged as a significant opportunistic pathogen, causing a range of skin disorders such as atopic dermatitis, dandruff, seborrhoeic dermatitis, and rosacea [2].

A defining feature of *S. epidermidis* is its remarkable capacity to form biofilms, which represents an important virulence factor. Biofilms are complex communities of bacterial cells embedded in a self-produced extracellular polymeric substance (EPS), which allow the bacteria to adhere to the surfaces of, for example, indwelling medical devices, and to resist external threats [3,4]. A biofilm not only enables *S. epidermidis* to effectively evade the host immune system but also significantly enhances its resistance to antimicrobial agents, posing serious challenges in clinical management. Biofilm-associated infections, such as catheter-related bloodstream infections, prosthetic joint infections, and prosthetic valve endocarditis, are particularly concerning due to their persistence and the frequent need for surgical removal of the infected device [5,6].

The ability of *S. epidermidis* to form biofilms and its adaptability to environmental pressures have contributed to its growing relevance in nosocomial infections. Furthermore, this bacterium has acquired a concerning array of antibiotic resistance mechanisms. Horizontal gene transfer (HGT) between *S. epidermidis* and other staphylococcal species, such as *S. aureus*, has facilitated the spread of resistance genes encoded on mobile genetic elements like the staphylococcal cassette chromosome (SCCmec) [7]. Methicillin-resistant *S. epidermidis* (MRSE) is currently identified in hospital environments, with resistance rates reported to exceed 70%. Resistance to methicillin is often linked with resistance to multiple antibiotic classes, significantly limiting available therapeutic options [2]. This multidrug resistance, combined with the protective nature of biofilms, has elevated *S. epidermidis* infections to critical concern in modern medicine.

Traditional antibiotics are often ineffective against biofilm-associated infections due to limited penetration into the biofilm matrix and the altered metabolic states of the bacteria within the biofilms, which reduce antibiotic susceptibility [8]. Consequently, the search for alternative therapeutic strategies has intensified in recent years. Natural products, with their diverse chemical structures and unique mechanisms of action, have garnered attention as promising candidates in the fight against antibiotic-resistant pathogens. In this context, the Celastraceae family has been widely used in traditional medicine in different parts of the world for the treatment of a variety of disorders. Different molecules isolated from these plants have shown a variety of activities such as in being antitumor, antioxidant, anti-inflammatory, stimulant, and, of course, antimicrobial, among others [9,10]. In our search for plant antimicrobial compounds, celastrol was isolated (Figure 1). It is a pentacyclic triterpenoid from the group of celastroloids, commonly found in the bark of the roots of Celastraceae species, and it has shown potent antibacterial activity [11,12,13,14,15,16,17].

Preliminary evaluations suggest that celastrol could act by interfering with multiple cellular pathways, including oxidative stress response and cell wall biosynthesis, making it a valuable candidate for combating bacterial pathogens resistant to standard antibiotics [11,16]. Some studies have also demonstrated that celastrol exhibits notable antibiofilm activity against various bacterial species. For example, celastrol has demonstrated potential in combating biofilm-associated and virulence-related infections caused by multidrug-resistant *Stenotrophomonas maltophilia* [18] and has also shown activity against anticariogenic bacterium *Streptococcus mutans* [19]. Moreover, its antibiofilm efficacy against staphylococcal and enterococcal strains has been highlighted in different studies targeting *Staphylococcus aureus* and *Enterococcus faecalis* [12,16,20,21]. While celastrol’s activity against other bacterial species has been investigated, its potential against *S. epidermidis* remains unexplored. The purpose of this work is to evaluate the antibacterial and antibiofilm activities of celastrol against this emerging opportunistic pathogen, focusing on elucidating its interference with biosinthetic pathways and biofilm formation mechanisms, thereby contributing to a deeper understanding of its potential antimicrobial applications.

## 2. Results and Discussion

### 2.1. Antibacterial Action of Celastrol Against S. epidermidis

In this study, the antibacterial activity of celastrol, in comparison with standard antibiotics targeting different bacterial pathways, was evaluated against *S. epidermidis*. Celastrol has demonstrated notable antibacterial activity against this bacterium, achieving a minimal inhibitory concentration (MIC) of 0.31 μg/mL and minimal bactericidal concentration (MBC) of 15 μg/mL [11]. These values were significantly lower than those of conventional antibiotics used as controls, including tetracycline, penicillin, vancomycin, and bacitracin (Table 1). In fact, the strain of *S. epidermidis* employed in this study demonstrated resistance to both penicillin and tetracycline [22], suggesting that celastrol may be more effective in inhibiting *S. epidermidis* than these antibiotics. This could be particularly advantageous in combating strains with varying levels of antibiotic resistance, as observed in numerous clinical isolates of *S. epidermidis* [23,24,25].

The activity of celastrol has also been demonstrated in previous studies against multidrug resistance strains such as methicillin-resistant *Staphylococcus aureus* (MRSA) [15,17,26], vancomycin-resistant enterococci (VRE) [12] or carbapenem-resistant *Klebsiella pneumoniae* when used in combination with thymol [13]. These findings underline celastrol’s antimicrobial potential, positioning it as a valuable complementary agent in antimicrobial therapy.

The time–kill curves of *S. epidermidis* exposed to celastrol at a concentration of 15 μg/mL showed an initial bacteriostatic effect, which transitioned to bactericidal activity after 24 h of treatment (Figure 2). This bactericidal effect was consistent across all inoculum sizes tested (from 10^3^ to 10^8^ CFU/mL), with activity maintained for up to 72 h, suggesting that celastrol exerts consistent antibacterial activity regardless of bacterial load. At 10^8^ CFU/mL, a notable decrease in optical density was recorded (Figure 3). This decrease indicates a lytic effect of celastrol on the bacterial culture, pointing to potential disruption of the bacterial membrane or cell integrity.

Yehia et al. [16] found that cells of *S. aureus* treated with celastrol were more susceptible to induced autolysis compared to untreated cells. This increase in autolytic activity was attributed to the negative regulation of the *msaB* gene, which is involved in the regulation of the cell death rate. In our study, the antimicrobial activity of celastrol was consistent across different growth phases. Specifically, celastrol added at the lag-phase exhibited a time–kill profile similar to that observed when added to actively growing cultures. This suggests that the efficacy of celastrol against *S. epidermidis* is not dependent on the bacterial growth phase, highlighting its potential utility for inhibiting both dormant and actively replicating bacterial populations. Additionally, the activity of celastrol added in lag-phase for 24 h exceeded that of standard antibiotics such as rifampicin. In the logarithmic growth phase, celastrol demonstrated a stronger bactericidal effect than rifampicin and similar to ciprofloxacin.

The effect of celastrol was further investigated by varying the concentration of the compound (3–25 µg/mL). During the first 6 h of treatment, all concentrations evaluated displayed a bacteriostatic effect, inhibiting bacterial growth without causing cell death. However, celastrol induced a bactericidal effect after 48 h of exposure

### 2.2. Effects of Celastrol on Macromolecular Synthesis and Solutes Uptake

To explore the potential mechanism of action against *S. epidermidis*, the effect of celastrol on the incorporation of radiolabeled precursors into the major biosynthetic pathways was investigated. Specifically, the incorporation of thymidine (DNA), uridine (RNA), leucine (proteins), and N-acetylglucosamine (cell wall) into their respective macromolecules was assessed. The incorporation of N-acetylglucosamine (NAG) into the cell wall was notably inhibited by over 50% within the first 2 min following the addition of celastrol, suggesting a rapid interference with cell wall biosynthesis (Figure 4). In contrast, the incorporation of thymidine, uridine, and leucine was progressively inhibited over a longer time course, with >50% inhibition occurring approximately 20–30 min. This differential pattern in inhibition suggests that celastrol predominantly targets cell wall biosynthesis of *S. epidermidis* during the initial stages of treatment, while its effects on nucleic acid and protein synthesis occur more gradually.

The uptake of biosynthetic precursors by *S. epidermidis* cells was also evaluated to determine whether the observed inhibition was due to a disruption in precursor entry into the cells. As shown in Figure 5, celastrol inhibited the uptake of NAG by more than 50% within 10 min of treatment, with no significant differences compared to thymidine. In contrast, the compound did not significantly reduce the uptake of the other biosynthetic precursors—thymidine, uridine, and leucine—by more than 50% during the 30 min evaluation period. This observation further supports the idea that celastrol may selectively interfere with cell wall biosynthesis while other metabolic processes remain less affected. Considering that the incorporation of NAG into peptidoglycan was inhibited within 2 min of treatment, it is unlikely that the primary effect of celastrol is on precursor uptake.

In previous research, our group demonstrated that celastrol inhibits macromolecular synthesis in *Bacillus subtilis*, likely through damage to the cytoplasmic membrane [11]. The arrest of solute uptake observed was closely followed by an immediate inhibition of macromolecular synthesis. By disrupting the transport of solutes and other essential molecules, celastrol appeared to compromise vital cellular functions, leading to broader antimicrobial effects in *B. subtilis*. The data obtained in the present study suggest that celastrol acts directly on the incorporation of precursors into the cell wall of *S. epidermidis*. The observed inhibition of uptake may be a secondary effect resulting from cellular stress or metabolic dysregulation caused by impaired peptidoglycan synthesis, which contributes to the bacteriolytic effect observed in previous experiments. The delayed inhibition of other biosynthetic processes further suggests that the antibacterial action of celastrol may involve additional interactions with multiple cellular targets, ultimately leading to bacterial cell death.

### 2.3. Effects of Celastrol on the Stability and Functionality of Bacterial Cell Membranes

The bacterial cytoplasmic membrane is a crucial structure that serves as a selective permeability barrier between the cytoplasm and the extracellular environment, regulating the passage of ions and molecules essential for bacterial survival. This membrane is a common target for various antimicrobial agents, as it prevents the loss of essential components such as ions, nucleotides, and other vital molecules. To further investigate the effects of celastrol on cell membranes of *S. epidermidis*, potential alterations in cell integrity and structural damage using a multifaceted approach were evaluated: a Live/Dead BacLight kit to evaluate membrane integrity; measurement of released cytoplasmic components (material absorbing at 260/280 nm and potassium-K^+^ leakage); and evaluation of oxygen uptake as a means of assessing membrane functionality. At a concentration of 15 μg/mL, celastrol did not induce significant membrane damage after 1 h of exposure, as evidenced by predominantly green fluorescence in treated populations (Figure 6). However, extended exposure (2 h) or higher concentrations of celastrol (30 μg/mL) resulted in a marked increase in red fluorescence, similar to the effect observed with clofoctol at 10 μg/mL used as positive control, indicating a loss of membrane integrity in a substantial proportion of the cell population.

Antibacterial agents that target the cytoplasmic membrane as their mechanism of action cause the rapid release of cytoplasmatic compounds from the cell, resulting from structural damage or disruption of the membrane’s permeability barrier. Consequently, the release of potassium or genetic material (DNA/RNA) is considered an early indicator of membrane damage. The measurement of the release of material absorbing at 260/280 nm, along with the quantification of cytoplasmic potassium leakage, demonstrated that celastrol did not induce a significant release of cytoplasmic components (only a slight release compared to the untreated control). This release was markedly lower than that induced by the positive control clofoctol, an antimicrobial whose mechanism of action primarily involves the inhibition of bacterial protein synthesis essential for maintaining membrane function, thereby indirectly destabilizing the cytoplasmic membrane integrity (Figure 7). We cannot rule out that the low release of cytoplasmic components after celastrol treatment could be due to the compound binding to the membrane, altering its function and permeability. During bacterial incubation with celastrol, the absorbance at 435 nm (the wavelength at which celastrol absorbs maximally in aqueous media) decreased in the supernatants after centrifugation, accompanied by a color shift in the sediment, suggesting celastrol’s adherence to the cell surface. This effect has been observed with netzahualcoyone, a quinone similar to celastrol, which is capable of attaching to the growing cells of *B. subtilis*, resulting in membrane alterations [27].

In addition to membrane integrity, the functional impact of celastrol on the cytoplasmic membrane was assessed by measuring oxygen consumption as an indicator of respiratory activity. Cellular respiration is a vital function of the cytoplasmic membrane, enabling energy production, maintaining intracellular redox balance, and providing protection against oxidative stress [28]. The electron transport chain, located within the membrane, is essential for aerobic respiration, with oxygen serving as the terminal electron acceptor to drive ATP synthesis. Impaired oxygen consumption is a characteristic feature of membrane-targeting agents, which interfere with the electron transport chain and ATP production, ultimately disrupting bacterial metabolism. When added at 15 μg/mL, celastrol was found to slightly inhibit oxygen consumption (<50%) in *S. epidermidis* during the 13 min experimental period (Appendix A). In contrast, a previous study with *B. subtilis* revealed a significantly different response, where celastrol induced an almost immediate cessation of oxygen consumption, comparable to the effect observed with sodium cyanide used as a positive control [11]. These results suggest that, although celastrol may compromise aerobic respiration and alter essential cytoplasmic membrane functions, the membrane is unlikely to be the primary target of action in *S. epidermidis.*

Further structural alterations induced by celastrol were visualized using transmission electron microscopy (TEM). After 1 h of treatment with celastrol, TEM images revealed changes in the cellular architecture of *S. epidermidis* compared to untreated controls. Among the observed alterations, mesosome-like structures were identified at the septa during cell division, suggesting that celastrol may interfere with normal division processes and contribute to structural anomalies within the bacterial cell envelope (Figure 8). These results suggest that while the celastrol primary action target appears to be cell wall synthesis, its action extends as a secondary effect to other cellular structures and functions. This effect also includes the cytoplasmic membrane, possibly as a consequence of structural instability or metabolic dysregulation, ultimately leading to a multifaceted disruption of bacterial homeostasis.

### 2.4. Action of Celastrol on Biofilm Formation and Pre-Formed Biofilms of S. epidermidis

The biofilm is essential for facilitating bacterial adhesion and providing structural support to the extracellular matrix, which is critical for its formation and long-term stability [29]. The biofilm creates a protective environment for the bacteria, making them highly resistant to antimicrobial therapies and immune defenses. *Staphylococcus epidermidis* is a prominent biofilm-forming pathogen and the most important species involved in medical device-related infections [30]. As a result, treatment often requires invasive interventions such as foreign body removal and prolonged antimicrobial therapy, which carry significant risks for patients and lead to higher healthcare costs [7]. This underscores the importance of exploring strategies aimed at preventing biofilm formation in this bacterial species. The antibiofilm activity of celastrol was evaluated by examining its impact on both the inhibition of biofilm formation and the disruption of pre-formed biofilms in *S. epidermidis*. For the inhibition of biofilm formation, celastrol was added at the beginning of the culture at concentrations ranging from 0.06 to 30 μg/mL. At a concentration of 0.94 μg/mL (3× MIC), a significant reduction in biofilm formation of over 70% was observed compared to untreated control with 0% inhibition/eradication (Figure 9). Scanning Electron Microscopy (SEM) images of these treated cultures (Figure 10C,D) show that celastrol inhibited the formation of the typical biofilm structure, with the cells appearing dispersed and failing to form tight clusters. However, no significant increase in biofilm inhibition was observed with concentrations exceeding 0.94 μg/mL, suggesting a plateau effect at concentrations just three times the MIC.

For the eradication of pre-formed biofilms, higher concentrations of celastrol were required. A concentration of 7.5 μg/mL was able to reduce the pre-formed biofilm by more than 70% (Figure 9). This treatment resulted in significant biofilm degradation, including the destruction of the extracellular matrix and detachment of cells from the surface (Figure 10E,F). These images contrast with the established biofilm observed in the untreated controls (Figure 10A,B). Concentrations higher than 7.5 μg/mL did not lead to a further significant improvement in biofilm eradication (72%), with only a modest increase in biofilm disruption (Figure 9). This indicates that celastrol is effective not only in preventing biofilm formation but also in disrupting and eradicating established biofilms in *S. epidermidis*. These findings align with previous studies, where celastrol has demonstrated significant antibiofilm activity across a variety of bacterial species. For instance, celastrol has been shown to reduce biofilm formation and metabolic activity in pre-existing biofilms of *S. maltophilia*, with a concentration-dependent effect that disrupts the extracellular matrix and decreases metabolic activity by over 80% [18]. Similarly, Li et al. reported that celastrol inhibited biofilm formation and viability within multi-species biofilms in oral microbiota by reducing the production of water-insoluble glucans, a key component of biofilm structure and adherence [19]. This antibiofilm activity effectively prevented the growth of pathogens such as *Streptococcus mutans*, suggesting that celastrol could play a significant role in maintaining oral microbial balance, potentially offering a novel approach for anticaries therapy. In another investigation carried out by Lu et al., celastrol also exhibited antibacterial and antibiofilm activity against enterococci, including vancomycin-resistant strains (VRE) [12].

Our data strongly suggest that celastrol exerts its antibacterial and antibiofilm effects by disrupting critical structural and metabolic components. This includes inhibiting NAG incorporation into the cell wall and destabilizing the extracellular matrix of biofilms, thereby interfering with biofilm formation, adhesion, and stability. These results position celastrol as a promising candidate for the development of new strategies to combat *S. epidermidis*-associated infections. While our study provides valuable insight into the antibacterial and antibiofilm effects of celastrol against *S. epidermidis*, the exact mode of action of celastrol remains unclear. Additional studies are needed to fully elucidate the molecular mechanisms behind its effects. Future investigations could focus on combined therapies, pairing celastrol with well-known antibiotics that have distinct mechanisms of action, to evaluate potential synergistic effects on biofilm inhibition or eradication. Moreover, molecular biology approaches could be employed to determine whether celastrol suppresses the expression of proteins involved in specific stages of biofilm formation.

## 3. Materials and Methods

### 3.1. Bacterial Strain and Culture Conditions

Cultures of *Staphylococcus epidermidis* (ATCC 14990 strain) were grown at 37 °C in nutrient broth (NB from Thermo Scientific^TM^ Oxoid, Basingstoke, UK). All culture media were sterilized by autoclaving at 121 °C for 20 min. For biofilm formation assays, NB media were supplemented with 1% glucose and were previously sterilized by filtration through 0.2 µm pore-size membrane filters (FP30/0.2 CA-S; Whatman, Maidstone, UK).

### 3.2. Celastrol and Other Antibacterial Compounds

Celastrol was isolated, purified, and characterized from the roots of Celastraceae species, following previously reported protocols [31,32]. The pure compound was dissolved in dimethyl sulfoxide (DMSO) prior to evaluation. Reference antibacterial agents, including ciprofloxacin, rifampicin, tetracycline, gentamicin, penicillin, vancomycin, bacitracin, and clofoctol (all from Sigma-Aldrich, St. Louis, MO, USA), were used as controls.

### 3.3. Antimicrobial Susceptibility Testing

The susceptibility of *S. epidermidis* to celastrol and control antibiotics was tested in triplicate using a broth microdilution method (concentration range: 0.06–60 μg/mL) in 96-well microtiter plates, following the M100 Clinical and Laboratory Standards Institute (CLSI) [22]. Wells containing the same DMSO concentration (not exceeding 1% *v*/*v*) served as controls. Initial microbial concentrations ranged from 1 to 5 × 10^5^ CFU/mL, and bacterial growth was tracked by measuring the increase in optical density at 550 nm (OD_550_) using a microplate reader (Infinite M200, Tecan, Männedorf, Switzerland) and by viable cell counts on nutrient agar plates. The minimal inhibitory concentration (MIC) was defined as the lowest compound concentration at which no microbial growth was observed relative to untreated controls. The minimal bactericidal concentration (MBC) was calculated by removing 100 μL from each well that did not exhibit visual bacterial growth and then subcultured on nutrient agar plates, followed by incubation at 37 °C for 24 h. The MBC was defined as the lowest concentration, and a ≥99.9% reduction of the initial inoculum was achieved.

### 3.4. Bacterial Growth Curves

#### 3.4.1. Bacterial Killing Assays

Overnight liquid cultures of *S. epidermidis* were diluted in Erlenmeyer flasks containing 10 mL of NB medium to achieve a working concentration of 1–5 × 10^5^ CFU/mL as previously described by Padilla-Montaño et al. [11]. Celastrol (15 µg/mL) was added either at time zero (lag phase) or after 4 h of incubation (log phase, OD_550_ ~0.4). Evaluations with known antibiotics or DMSO added in the same proportion served as positive and negative controls, respectively. Cultures were incubated at 37 °C on a rotatory shaker at 100 rpm, and bacterial growth was monitored hourly by measuring optical density and by colony counting on nutrient agar plates.

#### 3.4.2. Effect of Bacterial Inoculum Size and Celastrol Concentration

To assess the effect of bacterial inoculum size on celastrol activity, overnight liquid cultures of *S. epidermidis* were diluted in NB medium to achieve different inoculum concentrations (ranging from 10^3^ to 10^8^ CFU/mL). Celastrol (15 μg/mL) or the equivalent proportion of DMSO, used as a negative control, was added to each culture. To evaluate the effect of celastrol concentration on *S. epidermidis*, bacterial cultures (1–5 × 10^5^ CFU/mL) prepared as before were exposed to celastrol at different concentrations ranging from 3 to 25 μg/mL. Cultures were incubated at 37 °C in shaking, and growth was monitored at 3, 6, and 24 h by measuring optical density and counting colonies on nutrient agar plates, as described in our previous work [11].

### 3.5. Measurement of Radioactive Precursor Incorporation

Overnight cultures of *S. epidermidis* were diluted in Davis–Mingoli minimal medium [33] supplemented with 1% glucose, 0.1 g/L asparagine, and 2 g/L casamino acids (pH 7) to achieve a concentration of 10^6^ CFU/mL. Cultures were incubated at 37 °C in a rotary shaker at 100 rpm for about 3 h until an optical density (OD_550_) of 0.4 was reached. Aliquots of 10 mL were then transferred into pre-warmed flasks containing celastrol (15 µg/mL) along with one of the following labeled precursors: DNA synthesis ([6-^3^H] thymidine at 1 μCi/mL + 2 μg/mL unlabeled thymidine), RNA synthesis ([5-^3^H] uridine at 1 μCi/mL + 2 μg/mL unlabeled uridine), protein synthesis ([4,5-^3^H] leucine at 5 μCi/mL + 2 μg/mL unlabeled leucine), or cell wall peptidoglycan synthesis ([1-^14^C] N-acetyl-d-glucosamine at 0.1 μCi/mL). Samples were shaken at 37 °C, and at different time points, 0.5 mL aliquots were collected and treated with 2 mL of ice-cold 10% trichloroacetic acid (TCA). After 30 min on ice, samples were filtered onto glass microfiber filters (GF/C, Whatman Co., Maidstone, UK) and washed three times with 5 mL of ice-cold 10% TCA, followed by a 5 mL wash with 95% ethanol. Dried filters were then placed in vials containing scintillation cocktail and analyzed with an LKB Wallac Rackbeta counter (Perkin Elmer, Courtaboeuf, France). The synthesis of DNA, RNA, protein, and peptidoglycan was quantified by measuring the incorporation of radiolabeled thymidine, uridine, leucine, and N-acetylglucosamine (Amersham Biosciences Europe GmbH, Freiburg im Breisgau, Germany) into TCA-insoluble material. DMSO-treated cultures and specific pathway inhibitors served as negative and positive controls, respectively. Measurements were made in triplicate.

### 3.6. Measurement of Solutes Uptake

To measure solute uptake, total cell-associated radioactivity was quantified after the addition of celastrol to exponentially growing *S. epidermidis* cultures (OD_550_ ~0.2). Cells were grown in liquid medium at half-strength NB, transferred to pre-warmed flasks, and treated with celastrol (15 µg/mL) alongside radiolabeled precursors for DNA, RNA, protein, and peptidoglycan synthesis at the concentrations described above. At intervals over 30 min, 0.5 mL samples were collected, filtered through 0.45 μm Millipore filters (Millipore Corp., Bedford, MA, USA, type HA), and washed three times with 5 mL of phosphate buffer. Filters were then dried, and radioactivity was assessed as previously described. Parallel assays using identical DMSO volumes served as negative controls.

### 3.7. Integrity and Function of Cell Membrane

Bacterial membrane integrity was assessed by using the BacLight Live/Dead staining method employing propidium iodide (red fluorescence) to indicate membrane-compromised cells and Syto 9 (green fluorescence) to stain intact cells (catalog no. L-7012; Molecular Probes, Eugene, OR, USA), measurement of material release absorbing at 260/280 nm and potassium (K^+^) leakage. Log-phase *S. epidermidis* cultures (OD_550_ ~0.8) were centrifuged at 15,000× *g* for 10 min at 4 °C, followed by two washes with saline buffer. The resulting pellet was resuspended in saline buffer to reach a concentration of 1–2 × 10^8^ CFU/mL (or 1–2 × 10^7^ CFU/mL for K^+^ leakage experiments). Cultures were then exposed to celastrol (15 and 30 μg/mL) and incubated with shaking at 37 °C. Cultures treated with clofoctol (10 μg/mL) or the same proportion of DMSO were used as positive and negative controls, respectively. The BacLight assay was conducted in darkness for 20 min, following the manufacturer’s instructions, and cells collected after 60 and 120 min of treatment were observed under an epifluorescence microscope (Leica DM4B, Leica Microsystems GmbH, Wetzlar, Germany) equipped with a fluorescein-rhodamine filter at ×1000 magnification. Release of cytoplasmic material was monitored by measuring the optical density (OD_260_ and OD_280_) of the supernatant, following cell removal by centrifugation (9500× *g* for 10 min at 4 °C). Potassium (K^+^) release was quantified using an atomic absorption spectrophotometer (Model Thermo S-Series, Thermo Electron Corporation, Cambridge, UK). Additionally, the absorbance of celastrol at 435 nm (wavelength of maximal absorbance of celastrol in aqueous medium) was measured in saline buffer after removal of cells.

#### Oxygen Consumption

Suspensions of *S. epidermidis* in log phase of growth (OD_550_ ~0.8) were centrifuged, washed, and resuspended as described above to achieve a bacterial density of 1–2 × 10^7^ CFU/mL. Cell suspensions (2.7 mL) were supplemented with 0.3 mL of 10% glucose to measure oxygen uptake at room temperature, using a Clark oxygen electrode in a glass chamber with a magnetic stirrer. Celastrol (15 µg/mL) was introduced to the cell suspension, and steady-state oxygen levels were recorded after 4 min with a digital oxygen monitor (model YSI 5300, Yellow Springs, OH, USA). Parallel controls with DMSO and sodium cyanate (6.7 mM) were included as negative and positive controls, respectively.

### 3.8. Effect of Celastrol on Biofilm Formation

Anti-adherence activity assays of celastrol on *S. epidermidis* cells were conducted in 96-well flat-bottom microtiter plates following a modified protocol, according to Jardak et al. [34]. Celastrol (60 μg/mL) in NB medium supplemented with 1% glucose was added to the first column of wells and then serially two-fold diluted to obtain concentrations ranging from 30 to 0.06 μg/mL. An overnight culture of *S. epidermidis* was diluted in NB medium (1% glucose), and 100 μL was inoculated per well to achieve a working concentration of 1–5 × 10^5^ CFU/mL. Cultures without celastrol but with the same proportion of DMSO were used as negative control. Following a 24 h incubation at 37 °C, the NB and planktonic cells were discarded by inverting the plate, and wells were washed twice with 200 μL of sterile saline buffer (pH 7.2). Plates were dried at 60 °C for 30 min, then each well was stained with 150 μL of 0.2% crystal violet in 20% ethanol for 15 min. Excess stain was removed, and the attached bacteria were rinsed three times with water. To solubilize the biofilm, 200 μL of 33% (*v*/*v*) glacial acetic acid was added to each well and after 1 h of incubation at room temperature, the optical density was recorded at 570 nm with a microplate reader (Infinite M200, Tecan). Biofilm inhibition was calculated by comparing the absorbance of untreated and treated samples using the following formula:% biofilm inhibition = [(OD (growth control) − OD (sample))/OD (growth control)] × 100

### 3.9. Action of Celastrol on Pre-Formed Biofilm

The ability of celastrol to eradicate established biofilms was tested using 96-well flat-bottom microtiter plates [34]. Overnight cultures of *S. epidermidis* were prepared as previously explained. To initiate biofilm formation, 200 μL of the diluted bacterial culture was added to each well. After incubating the plates for 48 h at 37 °C, the NB medium and unattached cells were discarded by inverting the plates, and the wells were washed twice with sterile phosphate buffer. Celastrol solutions (ranging from 30 to 0.03 μg/mL in NB) were then added in 200 μL volumes to each well. Control wells were treated with the same amount of DMSO without celastrol. Following an additional overnight incubation at 37 °C, the wells were washed with phosphate buffer and stained with crystal violet as before, and absorbance was measured at 570 nm. The percentage of biofilm eradication was calculated as follows:% biofilm eradication = [(OD (growth control) − OD (sample))/OD (growth control)] × 100

### 3.10. Transmission Electron Microscopy

The impact of celastrol on *S. epidermidis* cells was examined via transmission electron microscopy (TEM). Bacterial cultures in the log phase (10^7^ CFU/mL) were exposed to celastrol (15 µg/mL) for 1 h at 37 °C, followed by centrifugation at 6500× *g* for 8 min at 4 °C. The collected pellets were rinsed with 0.1 M phosphate buffer (pH 7.2) and fixed in 2% OsO_4_ in buffer solution, then washed with distilled water. Semi-thin sections (1 µm) were prepared using a Reichert-Ultracut ultramicrotome and stained with toluidine blue; ultrathin sections were then counterstained with uranyl acetate and lead. Observations were performed on a Zeiss EM 912 transmission electron microscope (Carl Zeiss, Oberkochen, Germany), and images were captured using a Proscan Slow-scan CCD camera (ProScan GmbH, Lagerlechfeld, Germany) with EsiVision Pro 3.2 software (Soft Imaging System GmbH, Berlin, Germany). Control samples treated with the equivalent DMSO concentration were processed in parallel.

### 3.11. Scanning Electron Microscopy

Scanning electron microscopy (SEM) was employed to examine the architecture of *S. epidermidis* biofilms formed in the presence of celastrol and to assess the disruption of mature biofilms treated with the compound. The antiadherence properties of celastrol were evaluated on cultures initiated by preparing cell suspensions as previously described, followed by inoculation of 1.5 mL into 24-well microtiter plates in the presence of celastrol at 0.9 μg/mL, which were incubated at 37 °C for 24 h. To analyze biofilm eradication, the medium in pre-established biofilms was replaced with fresh medium containing celastrol at 7.5 μg/mL, and the plates were incubated for an additional 2 h. Subsequently, wells were gently washed twice with sterile saline buffer to eliminate planktonic cells. Afterward, biofilms were fixed in 2.5% glutaraldehyde prepared in 0.1 M phosphate buffer saline (pH 7.4) for 48 h. Samples were then washed twice with 0.1 M saline buffer (pH 7.4) for 10 min each. Post-fixation was carried out using 1% osmium tetroxide (OsO_4_) in 0.1 M PBS for 1 h, followed by two additional 10 min washes with 0.1 M PBS (pH 7.4). The samples were dehydrated in a graded ethanol series: 50%, 70%, 80%, 90%, and 100%, with each step lasting 10 min. For critical point drying, samples were sequentially treated with a mixture of ethanol 100% and hexamethyldisilazane (HMDS) in a 2:1 ratio for 10 min, followed by a 1:1 ratio for 10 min, and pure HMDS for another 10 min. After complete evaporation of the HMDS, biofilms were mounted onto aluminum stubs with double-sided carbon tape. The samples were then coated with a 15-nm gold layer using a sputter coating machine (Quorum Q150R S, Quorum Technologies, Lewes, UK). Imaging was performed using a Zeiss EVO LS-15 scanning electron microscope (Oxford Instruments, Abingdon, Oxfordshire, UK) operated at an accelerating voltage of 10 kV.

### 3.12. Statistical Analysis

Each evaluation was repeated across three independent experiments, and the means, along with standard deviations (SD), were calculated. To determine significant differences between treatments, a one-way analysis of variance (ANOVA) followed by Tukey’s post-hoc test (*p* < 0.05) was conducted for the measurement of radioactive precursor incorporation and solute uptake using R software, version 4.0.3 (R Foundation for Statistical Computing, Vienna, Austria).

## 4. Conclusions

This study highlights the significant antibacterial and antibiofilm activities of celastrol against *S. epidermidis*. Celastrol exhibited a potent antibacterial effect with a low MIC (0.31 μg/mL) and MBC (15 μg/mL), which outperforms several conventional antibiotics in terms of efficacy. Notably, its action was consistent across various bacterial growth phases, indicating its versatility in combating both dormant and actively proliferating bacterial populations. Additionally, celastrol demonstrated concentration-dependent activity, transitioning from bacteriostatic to bacteriolytic at higher concentrations.

The compound’s mechanism of action was linked to the inhibition of key biosynthetic pathways, particularly the disruption of cell wall synthesis, with an immediate inhibition of N-acetylglucosamine incorporation into the peptidoglycan layer. While celastrol also impacted other macromolecular synthesis processes, the disruption of cell wall integrity appeared to be its primary mode of action. Furthermore, the effects of celastrol extended to bacterial cell structures, including membrane integrity and division processes, as seen in TEM images.

Importantly, celastrol was effective in both preventing biofilm formation and disrupting pre-formed biofilms in *S. epidermidis*. The concentration-dependent activity of celastrol against biofilms, such as in significant biofilm eradication at concentrations of 7.5 μg/mL and higher, underscores its potential as an antibiofilm agent. This effect aligns with findings from other studies, which demonstrate celastrol’s broader antibiofilm efficacy against various bacterial species. The findings of this study could pave the way for future research into celastrol’s application as a therapeutic agent against multidrug-resistant staphylococcal infections, thus contributing to the development of alternative strategies in antimicrobial therapy.

## Figures and Tables

**Figure 1 antibiotics-14-00026-f001:**
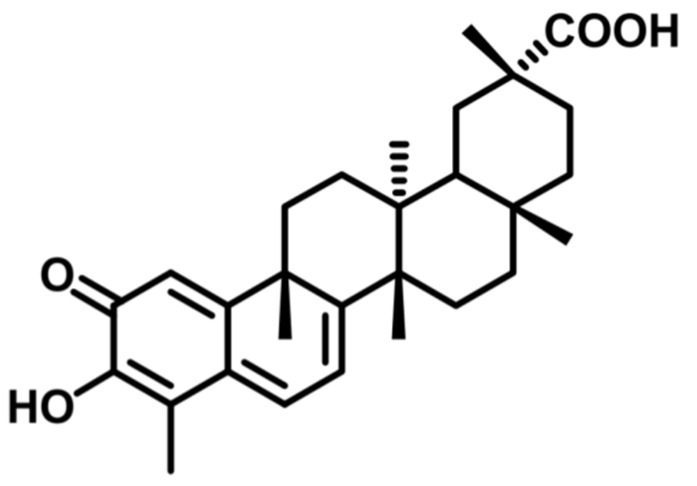
Chemical structure of celastrol.

**Figure 2 antibiotics-14-00026-f002:**
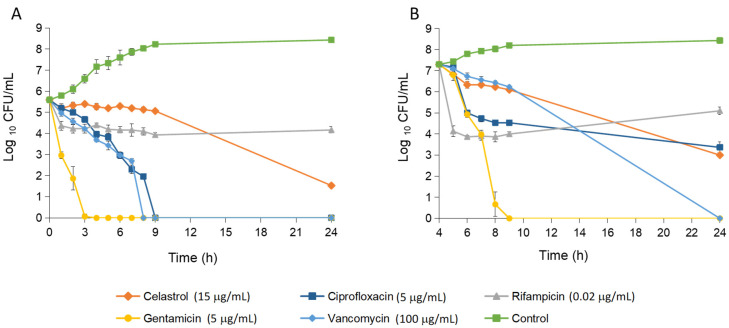
Killing curves of *S. epidermidis* in presence of celastrol or reference antibiotics. Untreated cell were used as a control. Compounds were added in lag-phase (**A**) and log-phase of growth after 4 h of preincubation (**B**). Error bars represent the standard deviation (SD) with *n* = 3.

**Figure 3 antibiotics-14-00026-f003:**
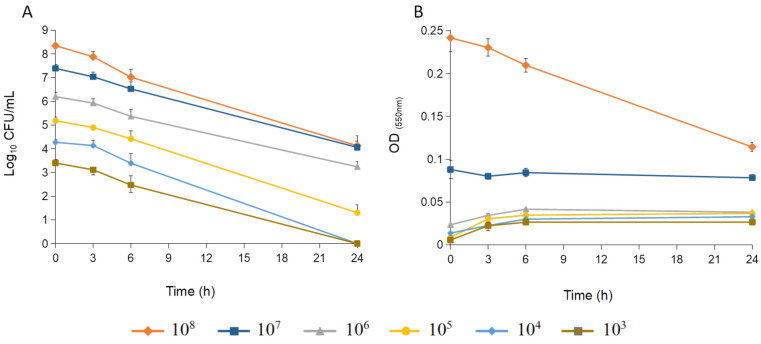
Effect of celastrol (15 μg/mL) on different inoculum sizes of *S. epidermidis* (ranging from 10^3^ to 10^8^ CFU/mL) at lag-phase of growth. Log_10_ of CFU counts (**A**) and optical density of the cultures (**B**). Error bars represent the standard deviation (SD) with *n* = 3.

**Figure 4 antibiotics-14-00026-f004:**
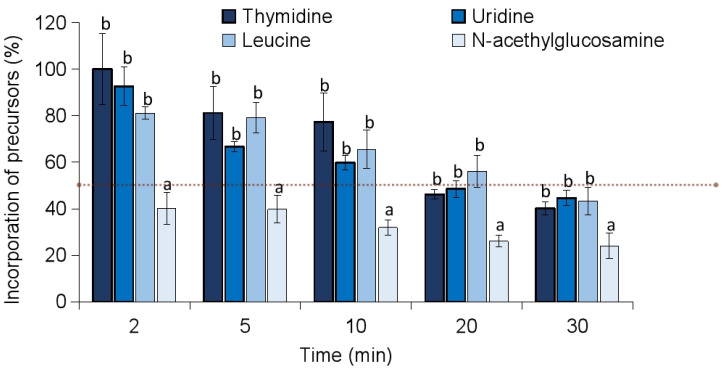
Effect of celastrol at 15 μg/mL in the incorporation of radiolabeled precursors [6-^3^H] thymidine, [5-^3^H] uridine, [4,5-^3^H] leucine and N-Acetyl-D-[1-^14^C] glucosamine for the synthesis of DNA, RNA, protein and cell wall, respectively, in *S. epidermidis*. Data are expressed as percentage (%) of precursors incorporated compared to controls without drugs but with the maximum proportion of DMSO (100% of incorporation). Red line indicates 50% of inhibition. Error bars represent the standard deviation (SD) with *n* = 3. Different letters above bars mean significant differences between treated cultures within each given time point (*p* < 0.05, one-way ANOVA; Tukey’s test).

**Figure 5 antibiotics-14-00026-f005:**
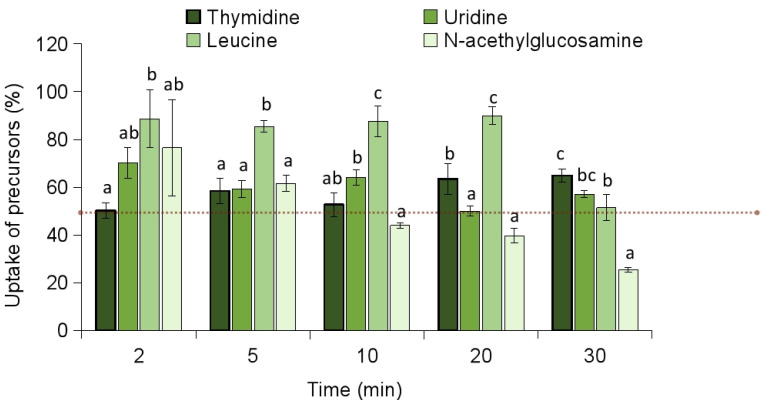
Effect of celastrol at 15 μg/mL in the uptake of radiolabeled precursors [6-^3^H] thymidine, [5-^3^H] uridine, [4,5-^3^H] leucine and N-Acetyl-D-[1-^14^C] glucosamine for the synthesis of DNA, RNA, protein and cell wall, respectively, in *S. epidermidis*. Data are expressed as percentage (%) of precursors uptaked compared to controls without drugs but with the maximum proportion of DMSO (100% of uptake). Red line indicates 50% of inhibition. Error bars represent the standard deviation (SD) with *n* = 3. Different letters above bars mean significant differences between treated cultures within each given time point (*p* < 0.05, one-way ANOVA; Tukey’s test).

**Figure 6 antibiotics-14-00026-f006:**
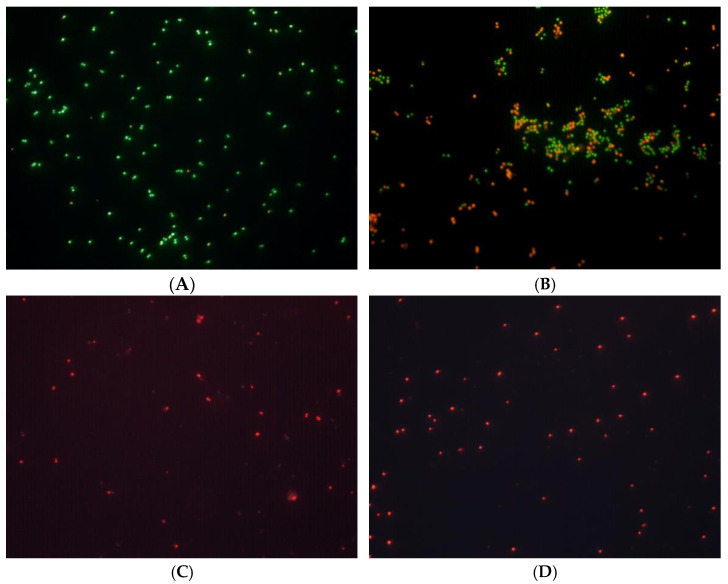
Epifluorescence microscopy images of *S. epidermidis* stained with propidium iodide and Syto 9 after treatment with celastrol at 15 μg/mL for 60 min (**A**) and 120 min (**B**) or 30 μg/mL for 60 min (**C**) and 120 min (**D**).

**Figure 7 antibiotics-14-00026-f007:**
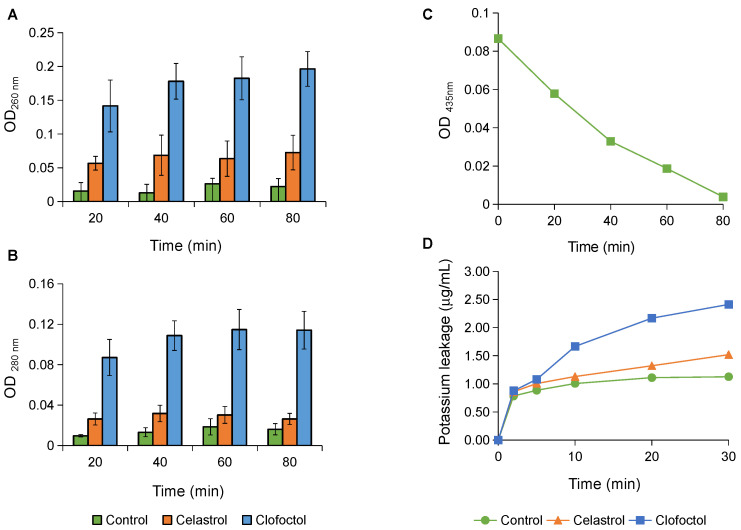
Effect of celastrol on the release of cytoplasmic components to the extracellular environment in *S. epidermidis.* Material absorbing at 260 nm (**A**) and 280 nm (**B**), and potassium release (**D**). Cell cultures treated with clofoctol served as positive control, while untreated cultures were used as negative control. Optical density at 435 nm (wavelength of maximal absorbance of celastrol) of cell suspension supernatant of *S. epidermidis* in saline buffer containing celastrol at 15 μg/mL (**C**). Error bars represent the standard deviation (SD) with *n* = 3.

**Figure 8 antibiotics-14-00026-f008:**
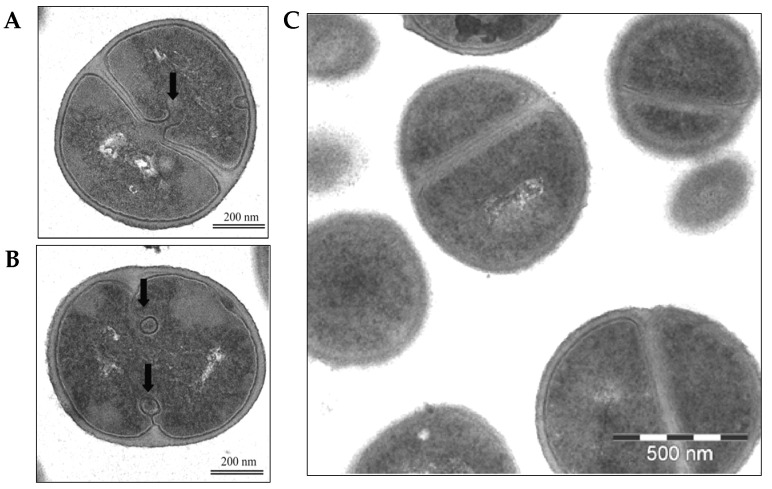
Transmission Electron Microscopy (TEM) of *S. epidermidis* cells treated with celastrol at 15 µg/mL for 1 h (**A**,**B**). Cell cultures without drugs but containing the maximum DMSO concentration were used as controls (**C**). Arrows indicate the presence of mesosome-like structures at the division septum site.

**Figure 9 antibiotics-14-00026-f009:**
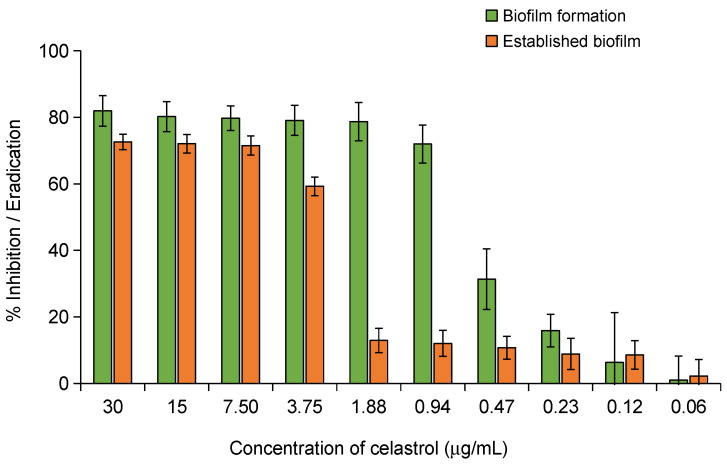
Effect of celastrol to inhibit biofilm formation or eradicate pre-formed biofilm of *S. epidermidis*. The percentage of inhibition or eradication is shown relative to the untreated control (not displayed), which represents 0% inhibition or eradication. The error bars represent standard deviation from the mean value. Error bars represent the standard deviation (SD) with *n* = 3.

**Figure 10 antibiotics-14-00026-f010:**
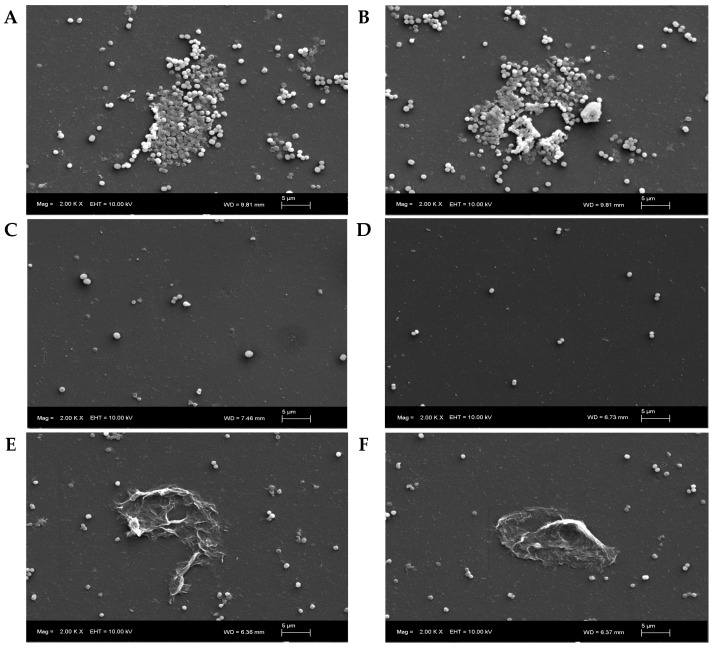
Representative Scanning Electron Microscopy (SEM) images of *S. epidermidis* biofilm. (**A**,**B**) show untreated control cells with early biofilm formation after 24 h of culture. (**C**,**D**) illustrate the anti-adherence activity of celastrol (0.94 μg/mL) added at the beginning of the culture, resulting in dispersed cells and the inhibition of biofilm formation after 24 h. (**E**,**F**) depict the effect of celastrol (7.5 μg/mL) after 24 h of treatment on a pre-formed biofilm, showing disruption and eradication of typical cell aggregates observed in untreated cultures.

**Table 1 antibiotics-14-00026-t001:** Minimal inhibitory concentration (MIC) and minimal bactericidal concentration (MBC) of celastrol, as well as other antibacterial agents used as control, against *S. epidermidis* ^1^.

Antibacterial Agents	MIC (μg/mL)	MBC (μg/mL)
Celastrol	0.312	15
Bacitracin	12.5	50
Ciprofloxacin	0.125	0.25
Clofoctol	1.25	12.5
Gentamicin	0.2	0.5
Penicillin	20	>40
Rifampicin	0.003	0.012
Tetracycline	>40	>40
Vancomycin	1.56	50

^1^ Values represent average obtained from a minimum of three experiments.

## Data Availability

The original contributions presented in this study are included in the article/Appendix A. Further inquiries can be directed to the corresponding author.

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
