# Peer review of "Interference of Celastrol with Cell Wall Synthesis and Biofilm Formation in Staphylococcus epidermidis"

_antibiotics, 2025, doi:10.3390/antibiotics14010026_

Round 1
Reviewer 1 Report
Comments and Suggestions for Authors
In this article titled "Interference of celastrol on cell wall synthesis and biofilm formation in Staphylococcus epidermidis" evaluates the antibacterial activity of celastrol in Comparison with standard antibiotics against S. epidermidis.
the authors utilized standardized methods for their analysis. Time-kill curves, macromolecular synthesis and cell membrane stability analysis were conducted by the authors to investigate the mechanism of celastrol action.
From their results, the authors showed that celastrol showcased significant antimicrobial activity against S. epidermis with an MIC of 0.31 μg/mL and MBC of 15 μg/mL respectively.
Furthermore, the authors conducted a time-course Epifluoresence microscopy on S. epidermidis after treatment with celastrol and showed
that extended exposure resulted in marked increase in red fluoresence, which is a standard for loss of membrane integrity in a substantial proportion of the cell population.
Analysis of celastrol on microbial biofilm is a unique method that evaluates antimicrobial activities against drug resistant biofilms from pathogenic organisms.
Statistical analysis was conducted in an appropriate and rigorous manner.
Author Response
Response to Reviewer #1
Dear Reviewer,
We would like to sincerely thank you for taking the time to review our manuscript titled "Interference of celastrol on cell wall synthesis and biofilm formation in Staphylococcus epidermidis." We greatly appreciate your thoughtful and thorough evaluation of our work.
Your positive feedback regarding as well as your acknowledgment of the statistical rigor and uniqueness of the biofilm studies, is both encouraging and gratifying.
It is particularly rewarding to know that you found our analysis of celastrol's antimicrobial activity, including its effects on biofilm formation and drug-resistant pathogens, to be valuable and well-conducted. We are delighted that our findings resonated with you and aligned with the goals of advancing knowledge in this field.
Once again, thank you for your valuable time and effort in reviewing our manuscript. Your feedback is highly motivating, and we are grateful for your support of our work.
Kind regards,
Laila Moujir
On behalf of all co-authors
Reviewer 2 Report
Comments and Suggestions for Authors
The research presented in the manuscript entitled “Interference of celastrol on cell wall synthesis and biofilm formation in Staphylococcus epidermidis” is interesting and quite well organized.
It is in your area of ​​study, as I saw, in the previous study, the antimicrobial action of celastrol and pristimerin was evaluated, and the mechanism of action against the spore-forming bacteria Bacillus subtilis was also approached.
A few suggestions were made to improve clarity: e.g. it would be appropriate to include more references in the sections describing the methods, etc.
Some corrections also need to be made: e.g. mL (in the graphs); values with dots, etc.

Author Response
Dear Reviwer,
We would like to express our sincere gratitude for your thoughtful and constructive comments on our manuscript entitled “Interference of celastrol on cell wall synthesis and biofilm formation in Staphylococcus epidermidis.” Your feedback has been extremely valuable, and we have carefully addressed each of your suggestions.
We have gone through your comments point by point and have made revisions and improvements throughout the manuscript based on your recommendations. These changes have significantly enhanced the clarity and quality of the text.
We believe these revisions have greatly improved the manuscript, and we are thankful for your input, which has helped us refine the work.
To facilitate your review, we have attached a PDF with your original comments along with our responses to each point (see the attachment). We hope that the revisions meet your expectations and that the manuscript is now in a more suitable form for publication.
Thank you again for your time and effort in reviewing our manuscript. We greatly appreciate your contributions.
Sincerely,
Laila Moujir
On behalf of all co-authors

Reviewer 3 Report
Comments and Suggestions for Authors
Dear Authors,
The manuscript titled “Interference of celastrol on cell wall synthesis and biofilm formation in Staphylococcus epidermidis” presents interesting and valuable findings regarding the potential of celastrol to affect the bacterial cell wall and biofilm formation. The study clearly outlines the mode of action of celastrol on S. epidermidis, which enhances the reader's understanding of its antibacterial mechanisms.
However, I would like to address a few points for clarification and revision:
Time-Killing Assay Results (Figure 2): The time-killing curves presented in Figure 2 show the effect of celastrol and known antibiotics on S. epidermidis. I noticed that rifampicin (0.02 µg/mL) did not inhibit bacterial growth at the 24-hour time point, whereas celastrol and other antibiotics maintained their inhibitory effects. Could you clarify why rifampicin loses its efficacy over time in this experiment? This observation requires further explanation.
Figure 9 (X-axis): The concentration values on the X-axis of Figure 9 are presented as "7,50", "3,75", and "1,88" µg/mL. These should be corrected to "7.50", "3.75", and "1.88" µg/mL for consistency and accuracy in numerical formatting.
Figure 10 (Panels E and F): The celastrol concentration in Panels E and F is indicated as "7,5 µg/mL". This should be corrected to "7.5 µg/mL" to match standard formatting.
Overall, the manuscript is well-structured, and the findings are significant in demonstrating the antibacterial potential of celastrol against S. epidermidis. With these minor revisions, the clarity and precision of the manuscript will be further improved.
Best regards
Author Response
Dear Reviewer,
We would like to sincerely thank you for the time and effort you dedicated to reviewing our manuscript and for your valuable comments and suggestions. We have carefully considered and addressed each of your observations, implementing the necessary clarifications and revisions (see our responses below).
We greatly appreciate your insights, which have undoubtedly contributed to improving the quality and clarity of our work. We hope that the modifications we have made meet your expectations and that the manuscript is now suitable for publication according to your criteria.
Thank you once again for your thoughtful feedback and constructive review.
Sincerely,
Laila Moujir
On behalf of all co-authors
Reviewer's comment:
Time-Killing Assay Results (Figure 2): The time-killing curves presented in Figure 2 show the effect of celastrol and known antibiotics on S. epidermidis. I noticed that rifampicin (0.02 µg/mL) did not inhibit bacterial growth at the 24-hour time point, whereas celastrol and other antibiotics maintained their inhibitory effects. Could you clarify why rifampicin loses its efficacy over time in this experiment? This observation requires further explanation.
Authors' response:
Thank you for your observation regarding the time-killing assay results and the behavior of rifampicin in our experiments.
Rifampicin is an antibiotic from the rifamycin class that, depending on its concentration and the specific experimental conditions, can act either as a bacteriostatic or bactericidal agent. At typical therapeutic concentrations, rifampicin primarily exhibits bactericidal activity by inhibiting bacterial RNA synthesis. It achieves this by binding to the beta subunit of DNA-dependent RNA polymerase, preventing the transcription of bacterial DNA into messenger RNA, which is lethal for the bacteria.
However, under certain conditions or at lower concentrations, rifampicin’s effect may shift to being bacteriostatic, meaning that it inhibits bacterial growth and multiplication without necessarily killing the cells. This behavior depends on factors such as the bacterial species, the growth phase of the bacteria, and the concentration of the drug used.
In our study, rifampicin demonstrated bacteriostatic activity when added during the lag phase of growth and bactericidal activity when added during the exponential growth phase. However, despite the initial inhibitory effects, bacterial regrowth was observed after 24 hours. This regrowth is likely due to the concentration of rifampicin used in the experiment. Specifically, rifampicin was applied at a concentration equivalent to 6 × MIC (0.018 µg/mL), whereas the other antibiotics used were tested at much higher multiples of their MICs: vancomycin at 64 × MIC, gentamicin at 25 × MIC, and ciprofloxacin at 40 × MIC.
This disparity in the relative concentrations of rifampicin compared to the other antibiotics likely explains its inability to sustain bactericidal activity over 24 hours when administered as a single dose.
We hope this clarifies the observed phenomenon, and we are grateful for the opportunity to further explain these results.
Reviewer's comment:
Figure 9 (X-axis): The concentration values on the X-axis of Figure 9 are presented as "7,50", "3,75", and "1,88" µg/mL. These should be corrected to "7.50", "3.75", and "1.88" µg/mL for consistency and accuracy in numerical formatting.
and
Figure 10 (Panels E and F): The celastrol concentration in Panels E and F is indicated as "7,5 µg/mL". This should be corrected to "7.5 µg/mL" to match standard formatting.
Authors' response:
Thank you for pointing out the inconsistencies in numerical formatting in Figures 9 and 10. We appreciate your attention to these details. We have carefully reviewed the figures and made the necessary corrections to ensure that the data are expressed in the standard format.
Reviewer 4 Report
Comments and Suggestions for Authors
Overview
I found the manuscript enjoyable to read. The title appropriately reflects both the methodology and outcomes of the study.
Regarding the first section on cell wall synthesis:
The manuscript begins by addressing the bacteriostatic and bactericidal effects of celastrol on Staphylococcus epidermidisacross different growth phases and inoculum sizes. This is followed by an exploration of celastrol's antibacterial mechanisms, focusing on its impact on the synthesis of DNA, RNA, proteins, and the bacterial cell wall. To further assess the effects of celastrol on these biosynthetic pathways, the uptake of precursors into the bacteria was evaluated. Based on the results, the manuscript suggests that celastrol inhibits the incorporation of NAG into the bacterial cell wall. While celastrol was shown to disrupt membrane integrity and induce cellular leakage, its effects were not as pronounced as those of the antibiotic used for comparison. The impact of celastrol on the cell wall was further examined using TEM, which revealed irregular cell envelope formation during bacterial cell division, as compared to the negative control.
In the second part of the manuscript, which focuses on biofilm formation:
The manuscript clearly discusses the inhibitory effects of celastrol on biofilm formation and its ability to eradicate pre-formed biofilm. These findings were further substantiated by SEM, providing additional evidence for celastrol's potential as an anti-biofilm agent.
As a non-native English speaker, I found the manuscript to be easily understandable, with no issues in comprehending the content. Overall, the manuscript appears scientifically sound and well-structured.
Title
Despite the existing title is well understood, the authors may consider to revise the title to the following:
Celastrol-Mediated Inhibition of Cell Wall Synthesis and Biofilm Formation in Staphylococcus epidermidis
The proposed new title is more direct and precise as it clearly states that celastrol inhibits of both cell wall synthesis and biofilm formation. "Celastrol-mediated" highlighting that celastrol is actively involved or facilitating the inhibition. "Interference" might imply a less direct or less specific interaction, which is not as clear as "inhibition” which is more precise for a scientific manuscript.
Introduction
- Suggest including a diagram of the plant from which celastrol is isolated.
- Line 96-98: Suggest revising the purpose or the aim to be more specific and concise, to including the mechanism.
Results and Discussion
- Justify the selection of these antibiotics for MIC and MBC testing to ensure the comparison in the discussion is relevant and well-supported.
- Justify why only few of the antibiotics used in MIC and MBC were tested in the time-kill curves and why with such concentrations? As the concentration of celastrol used based on the obtained MBC.
- Line 133-134: Is this statement a hypothesis or supported by any previous study?
- Section 2.2: Justify why antibiotic that can inhibit synthesis of DNA, RNA, proteins and cell wall was not used as the respective positive control, in the two assays.
- Section 2.3: Justify why a positive control was not used to assess the membrane integrity, but a positive control was used in to assess the cytoplasmic membrane?
- Line 288-289: Suggest revising the mechanism of clofoctol as this agent does not disrupt the cytoplasmic membrane directly.
- Line 203, 244, 599: Revise P<0.05 to p<0.05
- Line 418: Revise 7,5 to 7.5.
- Since the mechanism of antibiofilm properties was not conducted, suggest discussing more about the mechanisms might be involved based on previous studies.
- Line 420-426: Recommend enhancing this section by elaborating on the study's limitations and suggesting specific assays that could be conducted to further investigate this topic.
Materials and Methods
- Line 457: Is it 3 h or 4 h (Line 146)?
- Line 505: K+
- Line 506: 15,000x g
- Line 579: 37 °C
- Line 586: 0.1 M
- Section 3.12: Specify which assays that you did statistical analyses.
Conclusions
Well written.
Author Response
Reviewer's comment:
Title
Despite the existing title is well understood, the authors may consider to revise the title to the following:
Celastrol-Mediated Inhibition of Cell Wall Synthesis and Biofilm Formation in Staphylococcus epidermidis
The proposed new title is more direct and precise as it clearly states that celastrol inhibits of both cell wall synthesis and biofilm formation. "Celastrol-mediated" highlighting that celastrol is actively involved or facilitating the inhibition. "Interference" might imply a less direct or less specific interaction, which is not as clear as "inhibition” which is more precise for a scientific manuscript.
Authors' response:
We appreciate your thoughtful suggestion regarding the title of the manuscript. While we agree that "Celastrol-Mediated Inhibition" provides a direct and precise description of the effects observed, we believe that "Interference" better reflects the broader scope of celastrol's activity demonstrated in our study. Our findings suggest that celastrol not only inhibits cell wall synthesis and biofilm formation but may also interfere with other cellular processes, which we plan to explore further in future research. For this reason, we would prefer to retain the original title, as it encapsulates the potential multifaceted mechanisms of celastrol's action.
However, if you still believe that the suggested title would improve the clarity and impact of the manuscript, we are open to adopting it.
Reviewer's comment:
Introduction
- Suggest including a diagram of the plant from which celastrol is isolated.
Authors' Response:
Thank you for your suggestion to include a diagram of the plant from which celastrol is isolated. We appreciate your valuable feedback.
In the Materials and Methods section of our manuscript, we have already referenced two studies that detail the process of obtaining celastrol, including the source plant and the methodology for its isolation. These references provide a comprehensive explanation of the raw material and the procedures used.
While we understand the value of including such a diagram, we believe that it would not add significant originality or novelty to our work, as it would necessarily rely on previously published material. Incorporating this type of figure might also present potential copyright or third-party conflicts, as it would need to be based on existing published diagrams or descriptions.
For these reasons, we consider that our current approach sufficiently addresses this aspect while maintaining the focus and originality of our manuscript.
Reviewer's comment:
- Line 96-98: Suggest revising the purpose or the aim to be more specific and concise, to including the mechanism.
Authors' Response:
Thank you for your insightful comment regarding the purpose statement in lines 96–98. We agree that a more specific and concise description that includes the mechanism would enhance the clarity and precision of this section. The revised text now includes a clearer reference to the mechanisms involved in celastrol's activity, particularly its interference with biosynthetic pathways and biofilm formation. We hope this revision meets your expectations and improves the clarity of our manuscript.
Results and Discussion
Reviewer's comment:
- Justify the selection of these antibiotics for MIC and MBC testing to ensure the comparison in the discussion is relevant and well-supported.
Authors' Response:
We selected a range of antibiotics with different mechanisms of action to perform MIC and MBC testing against our S. epidermidis strain in order to provide a broad comparison of antibacterial efficacy. By including antibiotics with varying modes of action, we were able to assess the effectiveness of celastrol relative to established treatments and evaluate whether it exhibited superior activity against this particular strain. Additionally, this comparison was essential to establish appropriate concentrations of the antibiotics for subsequent experiments, such as Time-Kill assays, ensuring that our experimental design was robust and relevant.
We have added a clarification at the beginning of the Results and Discussion section to indicate that the standard antibiotics used in this study target different bacterial pathways, which justifies their inclusion for comparison with celastrol.
Reviewer's comment:
- Justify why only few of the antibiotics used in MIC and MBC were tested in the time-kill curves and why with such concentrations? As the concentration of celastrol used based on the obtained MBC.
Authors' Response:
We selected a limited number of antibiotics for the time-kill assays based on their MIC and MBC values and their relevance to the study. Antibiotics with high MBC values (>10 µg/mL), such as penicillin (>40 µg/mL), bacitracin (50 µg/mL), and clofoctol (12.5 µg/mL), were excluded from the time-kill assay as their concentrations would not provide meaningful comparative data in this context.
However, we included vancomycin in the time-kill assay to ensure a broader range of antibiotics with different mechanisms of action. Vancomycin, despite its higher MBC (50 µg/mL), targets bacterial cell wall synthesis, which complements the evaluation of celastrol’s effect on the same target. By including vancomycin, we aimed to provide a more comprehensive comparison of celastrol’s antibacterial activity against antibiotics with diverse mechanisms of action. In addition, vancomycin is the alternative antibiotic for the treatment of infections caused by methicillin-resistant S. epidermidis (MRSE).
Concentrations above the MIC were always used for the study of growth curves. These concentrations were determined based on the antimicrobial's mechanism of action (bacteriostatic or bactericidal). For instance, ciprofloxacin, with an MIC of 0.125 and an MBC of 0.25, and exhibiting bactericidal activity, was used at 4 times the MIC value. In the case of rifampicin, which demonstrated a bacteriostatic mechanism under the tested conditions (MIC of 0.003 and MBC of 0.012), it was tested at 6 times the MIC. The same approach was applied to the other antibiotics.
Reviewer's comment:
- Line 133-134: Is this statement a hypothesis or supported by any previous study?
Authors' Response:
The statement in lines 133-134 is based on the data obtained in our study and is not supported by previous studies. We suggest that celastrol exerts consistent antibacterial activity regardless of bacterial load based on the observations made during our experiments, where the bactericidal effect was maintained across all inoculum sizes tested.
Reviewer's comment:
- Section 2.2: Justify why antibiotic that can inhibit synthesis of DNA, RNA, proteins and cell wall was not used as the respective positive control, in the two assays.
Authors' Response:
In our study, we did perform assays using specific inhibitors for each biosynthetic pathway (DNA, RNA, proteins, and cell wall synthesis), and we have the corresponding data. However, we decided not to include them in the manuscript to simplify the presentation of the figures. As shown, the figures illustrate the effect of celastrol on the incorporation and uptake of the four biosynthetic precursors over time, allowing us to observe which pathway was affected first. Including additional controls would have required separate graphs for each precursor (a total of 8 graphs across the two assays), which we felt would complicate the presentation and not provide additional relevant information. However, if you deem it necessary, we are happy to include these controls in additional figures for a revised version of the manuscript.
Reviewer's comment:
- Section 2.3: Justify why a positive control was not used to assess the membrane integrity, but a positive control was used in to assess the cytoplasmic membrane?
Authors' Response:
We would like to clarify that we did use clofoctol as a positive control as mentioned in the Materials and Methods section. However, this was not explicitly mentioned in the Results section. To address this, we have now included this detail in the Results section as well, ensuring that it is clear the positive control was used, even though the data were not presented in the figure for simplicity and to focus on the key findings.
Reviewer's comment:
- Line 288-289: Suggest revising the mechanism of clofoctol as this agent does not disrupt the cytoplasmic membrane directly.
Authors' Response:
Thank you for pointing this out. Indeed, the mechanism of action of clofoctol does not directly involve the disruption of the cytoplasmic membrane but is primarily due to the inhibition of the synthesis of proteins essential for maintaining membrane function, thereby indirectly destabilizing the cytoplasmic membrane integrity.
To clarify this in the text, we have revised the description of clofoctol's mechanism of action, specifying that its effect on the cytoplasmic membrane is secondary. The revised text now reflects this detail more accurately.
Reviewer's comment:
- Line 203, 244, 599: Revise P<0.05 to p<0.05
Authors' Response:
Done.
Reviewer's comment:
- Line 418: Revise 7,5 to 7.5.
Authors' Response:
Done.
Reviewer's comment:
- Since the mechanism of antibiofilm properties was not conducted, suggest discussing more about the mechanisms might be involved based on previous studies.
Authors' Response:
We appreciate the suggestion to further discuss the mechanisms underlying celastrol’s antibiofilm activity. While the precise molecular mechanisms of celastrol’s antibiofilm properties were not directly investigated in our study, we have based our discussion on previous studies that suggest potential mechanisms. Celastrol has been shown to affect biofilm formation and disruption through various pathways, including the inhibition of extracellular matrix production and disruption of the biofilm’s structural integrity. In line with previous research, we hypothesize that celastrol may interfere with key processes involved in biofilm formation, such as the production of extracellular polymeric substances.
It is important to note that there are still relatively few studies that investigate the antibiofilm effect of celastrol on S. epidermidis. Most of the existing research on celastrol's antibiofilm activity has focused on other bacterial species, with mechanisms such as disruption of the extracellular matrix, reduction in metabolic activity, and interference with cell signaling being suggested as contributing factors. However, the limited number of studies on S. epidermidis means that further research is needed to fully elucidate the precise mechanisms of action in this specific pathogen.
We have already included references to these studies in our discussion to provide additional context for the potential mechanisms involved.
Reviewer's comment:
- Line 420-426: Recommend enhancing this section by elaborating on the study's limitations and suggesting specific assays that could be conducted to further investigate this topic.
Authors' Response:
Thank you for your valuable suggestion. We have revised the conclusion of the discussion to provide a more balanced perspective, beginning with an acknowledgment of the study's limitations, followed by a detailed summary of the key findings. We have included potential avenues for further research at the end of the discussion section which could help elucidate the precise mechanisms of action of celastrol. We hope this revision addresses your recommendation and enhances the overall clarity of the discussion.
Reviewer's comment:
Materials and Methods
- Line 457: Is it 3 h or 4 h (Line 146)?
Authors' Response:
Thank you for pointing out this discrepancy. You are correct that celastrol was added during the exponential phase of growth at 4 h, not 3 h as originally stated. We have corrected this error in the Materials and Methods section to accurately reflect the timing of treatment.
Reviewer's comment:
- Line 505: K+
- Line 506: 15,000x g
- Line 579: 37 °C
- Line 586: 0.1 M
Authors' Response:
We have incorporated all the suggested corrections in the new version of the manuscript.
Reviewer's comment:
- Section 3.12: Specify which assays that you did statistical analyses.
Authors' Response:
We have specified in section 3.12 that the statistical analysis was conducted specifically for the measurements of radioactive precursor incorporation and solute uptake. We hope this clarifies the scope of the statistical analysis in our study.
